# SHAVE PEAKS, DON'T FILL VALLEYS: UPPER-TAIL RISK BALANCING IMPROVES ROBUSTNESS WITHOUT ACCURACY LOSS

## ABSTRACT

Many sequence models achieve strong average performance yet exhibit **concentrated internal dependencies**: removing just a few "critical units/time positions" causes disproportionate degradation. We propose **RBRL (Risk-Balanced Representation Learning)**, which applies financial risk allocation principles to neural network training by constraining **attribution concentration** through adaptive risk budgets. RBRL uses a stable attribution signal (AEC: activation $\times$ gradient with EMA normalization) and imposes upper-tail constraints via quantile budgets and soft-Top-K penalties, enabling "peak-shaving" without compromising main objectives through dual-only training that preserves backbone gradients. Across S&P 500 and ETT datasets, RBRL **improves robustness under a tunable computational overhead while maintaining baseline-level accuracy on S&P 500**; on ETT, RMSE changes show mixed results across subsets; on S&P 500, differences are small but not statistically significant (RMSE $p$=0.216; MAE $p$=0.201; directional accuracy unchanged). Our comprehensive evaluation across 68 configurations demonstrates architecture-agnostic applicability to LSTM, iTransformer, and other sequence models. We position this as a **robust reliance training paradigm**: proactively dispersing dependencies during training rather than addressing brittleness post-hoc.

## 1 INTRODUCTION

Contemporary sequence models often concentrate prediction reliance on a few units or time positions: masking these "peaks" causes disproportionate errors. We introduce Risk-Balanced Representation Learning (RBRL), which uses a stable attribution signal (AEC: activation $\times$ gradient with EMA normalization) and applies upper-tail controls via quantile budgets and soft-Top-K, shaving peaks while keeping predictive loss central. We formalize reliance concentration as a non-negative distribution over hidden unit$\times$time and track inequality with Gini/HHI/Top-K. Empirically, RBRL keeps S&P 500 accuracy at baseline level (differences not statistically significant) and shows mixed RMSE changes across ETT subsets, while consistently reducing concentration and brittleness under targeted masking.

Mechanistically, RBRL allocates adaptive budgets to units/time and applies symmetric penalties for excess/shortage (peak-shaving with light valley-filling) with dual updates; a dual-only mode preserves backbone gradients, and an end-to-end path is available when needed. These choices yield a stable, low-variance signal and a tunable compute tax without inference overhead.

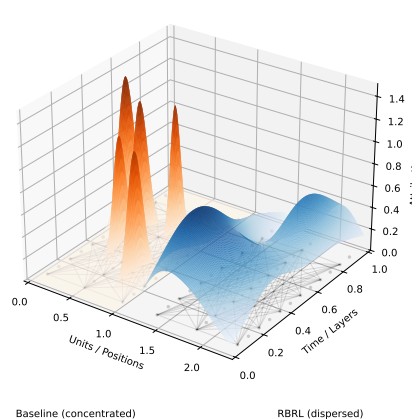

Baseline (concentrated)      RBRL (dispersed)

Figure 1: Attribution distribution: RBRL transforms concentrated "peaks" (left) into dispersed, balanced reliance (right; *illustrative surface for intuition*).

We position RBRL as a **robust reliance paradigm**: it's not another regularization trick, but elevating "**controlling reliance destinations**" to first-class status. The approach is backbone-agnostic (LSTM, Mamba, Transformer all supported via wrapper), inference-cost-free, achieving "risk shaving from peaks and reasonable spreading" purely through training-time budgets and gating.

**Contributions.** (1) **Problem formalization.** We operationalize internal reliance concentration over hidden units/time using a stable AEC signal and inequality indices (Gini/HHI/Top-K). (2) **Training principle.** RBRL combines quantile budgets and soft-Top-K with a dual schedule to control attribution tails without directly optimizing the indices. (3) **Evidence.** Across S&P 500 and ETT, accuracy is baseline-level on S&P 500 and mixed on ETT, while concentration decreases and targeted-mask brittleness is reduced.

## 2 RELATED WORK

### 2.1 ROBUST TRAINING AND REGULARIZATION

A substantial body of work has explored ways to improve generalization and robustness by augmenting the standard predictive loss. Weight decay and dropout remain widely used techniques, but they primarily affect parameter norms or co-adaptation and do not explicitly control the model's reliance on particular features or hidden units. More recent methods regularize the input–output Jacobian to reduce sensitivity: Ross et al. (2017a) introduced Right for the Right Reasons (RRR), which penalizes input gradients to enforce explanations consistent with domain knowledge or to diversify decision boundaries. Adversarial training and its variants (Madry et al., 2018) remain the de facto standard for robustness, yet they tend to increase computational cost and often degrade clean accuracy. Our Risk-Balanced Representation Learning (RBRL) differs by using the attribution distribution as a surrogate for concentration risk and applying *upper-tail controls* to it rather than suppressing all gradient norms.

### 2.2 INTERPRETABILITY AND ATTRIBUTION

A parallel literature focuses on interpreting neural networks via post-hoc attribution. Gradient-based methods such as Gradient × Input, Integrated Gradients (IG) and SmoothGrad attribute a model's prediction to their input features; these methods satisfy desirable axioms like sensitivity and implementation invariance, but they are typically used for *explanation* rather than training. Sundararajan et al. (2017) formalized axioms for attribution and proposed IG as a principled method requiring only standard gradient calls, while Smilkov et al. (2017) showed that averaging noisy gradients sharpens sensitivity maps (SmoothGrad). However, these attributions are usually computed *after* training and do not influence the learned representation. Nevertheless, these works operate at the level of input features or attention weights, not on hidden units or time positions. Our work takes an **orthogonal perspective**: rather than aligning explanations with annotations, we monitor and disperse the attribution distribution over hidden units and time steps, regardless of whether the attributions correspond to correct causal features.

### 2.3 ATTENTION ENTROPY AND UNIFORMITY

Transformers and other attention-based architectures exhibit a phenomenon where attention scores collapse to a few positions, leading to training instabilities. Zhai et al. (2023) empirically tracked the entropy of attention heads and observed that low attention entropy correlates with oscillating losses; they called this entropy collapse and proposed $\sigma$-Reparam, a spectral-normalization trick to prevent it. Other concurrent work adds penalties to encourage more uniform attention. While these methods explicitly regularize attention distributions, they focus on the *input-level* attention matrix and are limited to attention-based models. In contrast, RBRL is architecture-agnostic and applies dispersion principles to any sequence model by measuring gradient-based attributions over hidden units and time positions.

## 2.4 FAIRNESS, RESOURCE ALLOCATION, AND INEQUALITY METRICS

Our formulation draws inspiration from inequality measurement in economics. The Gini coefficient[1] (Gini, 1912) is a measure of statistical dispersion used to quantify income or wealth inequality; it ranges from 0 (perfect equality) to 1 (maximal inequality) and is defined in terms of the Lorenz curve. The Herfindahl–Hirschman Index (HHI) (Herfindahl, 1950), originally from industrial organization, measures market concentration by summing squared market shares; higher values indicate less competition. Such indices have been repurposed to evaluate the fairness of resource allocation in health care, education, and ecology. Our work is the first, to our knowledge, to apply these inequality indices to the *attribution distribution* of neural networks, interpreting high concentration as a form of reliance risk. We also consider Top-K mass, the fraction of attribution carried by the largest K% of units/time steps, akin to concentration ratios used in economics. Unlike existing fairness-oriented learning (e.g., ensuring equalized odds across demographic groups), our use of inequality metrics is internal: it aims to disperse reliance across a model's computational pathways without external annotations or demographic labels.

## 3 METHODOLOGY

### 3.1 ATTRIBUTION-ENHANCED CONTRIBUTIONS (AEC)

We measure the model's internal reliance by Attribution-Enhanced Contributions (AEC)—the element-wise product of a unit's activation and the gradient of the loss w.r.t. that activation, taken in absolute value:

$$\text{AEC}(h) \;=\; \big| \, h \odot \nabla_h \mathcal{L} \, \big|.$$

Implementation-wise, we compute gradients only for *activations* (not parameters) captured via forward hooks, then form |activation × grad|, and apply per-layer normalization using running statistics to stabilize scale drift across training. Finally, to aggregate AEC for analysis and budgeting, the code averages over batch and (optionally) time to obtain unit-level and time-level attribution profiles per layer.

### 3.2 RISK-BALANCED REPRESENTATION LEARNING (RBRL)

RBRL adds a *risk-concentration* regularizer to the prediction loss, shaping the internal attribution distribution so that the model does not over-rely on a few units or timesteps.

#### 3.2.1 BUDGETS AND SYMMETRIC "SHAVE-PEAKS/FILL-VALLEYS" PENALTY

For each layer we maintain an adaptive attribution budget $\rho$ (EMA/quantile-based) and a non-negative dual weight $\lambda$. Let $a$ denotes the unit-level AEC (averaged over batch and time). The penalty per unit is

$$\lambda\big[(a - \rho)_+^2 \;+\; \alpha_{\text{excess}}(\rho - a)_+^2\big],$$

which discourages excess above the budget while encouraging under-utilized units to participate (symmetric "shave-peaks/fill-valleys"). The code implements this symmetric form and accumulates it across units; per-time penalties are defined analogously (Sec. 3.2.3).

Dual variables are updated by gradient ascent on the excess $(a - \rho)$, with clamping to avoid divergence; time-level duals are updated similarly.

---

[1]Unless otherwise specified, Gini is computed at the **unit level** and aggregated across layers using **AEC** with **EMA** normalization (Sec. 4). Time-level or layer-wise Gini, when reported, is labeled explicitly and **not** directly comparable to unit-level Gini.

### 3.2.2  Soft-Top-K concentration control

To directly control tail concentration, we optionally replace the mean aggregator with a soft-Top-K operator that emphasizes the largest attributions (temperature-controlled). This operator is used both at unit- and time-levels when enabled.

### 3.2.3  Time-dimension control

When the AEC tensor is 3-D $(B, T, H)$, RBRL can penalize temporal concentration by computing time-wise aggregations (mean or soft-Top-K over units) and applying the same symmetric budget penalty w.r.t. a time-budget $\rho_{\text{time}}$.

### 3.2.4  Adaptive budgets

Budgets $\rho$ (and $\rho_{\text{time}}$) are adaptive: the implementation updates them from observed AEC using EMA/quantile rules to track the tail while avoiding instability. (See `update_adaptive_budget`.)

### 3.2.5  Training modes, PCGrad, and gradient gating

RBRL supports two modes. In dual-only, AEC is detached, and the risk penalty updates only the dual/budget processes; the network sees only the prediction loss. In end-to-end, the penalty participates in backprop through AEC (with the e2e surrogate to keep graphs safe). The trainer selects the mode per batch, sums loss and risk penalty when e2e, and applies PCGrad (Yu et al., 2020) when both objectives have conflicting gradients.

Additionally, gradient gating reduces gradients for units whose current AEC violates budgets the most; violations are mapped to an exponential gate and applied to LSTM parameters.

At the end of each epoch, we compute the average AEC, update duals, and budgets, store budget history, and log concentration metrics for monitoring.

## 4  Experimental Setup

**Datasets.** We evaluate on two time series datasets: (1) ETT dataset (Zhou et al., 2021) (Electricity Transformer Temperature, hourly) using the standard train/val/test split, and (2) S&P 500 financial time series for main validation. Our comprehensive ablation study spans 68 configurations across 5 random seeds, totalling 340+ experiments.

**Models and training.** Unless stated otherwise, we use a 2-layer LSTM (Hochreiter & Schmidhuber, 1997) (hidden size 64, dropout 0.2) optimized with Adam (Kingma & Ba, 2015) and early stopping. RBRL is enabled with dual-only or end-to-end modes, adaptive budgets (quantile or EMA), optional time control (weight = 0.5), and soft-Top-K where specified. Representative configs are recorded in the ETT dataset runs (e.g., `training_mode: dual_only`, `lambda_rp: 0.02`, `budget_method: quantile (q=0.90/0.95)`).

**Evaluation protocol.** We evaluate across Phase 1 (S&P 500, n=5 seeds), Phase 2 (ETT datasets, n=3 seeds), and Phase 6 (adversarial robustness, n=1 per method). All results use consistent preprocessing and 70%/15%/15% splits, with Bonferroni-corrected statistical testing. We report RMSE/MAE, concentration metrics (Gini/HHI/Top-K), and targeted-masking robustness. Gini coefficients are computed over unit-level attributions aggregated across layers using AEC with EMA normalization. Additional metrics: Directional accuracy remains unchanged (Baseline 49.67%±1.70% → RBRL 49.33%±1.89%, $p$=0.423). All figures show 95% CIs via bootstrap resampling. Detailed statistical procedures are provided in Appendix A.3.

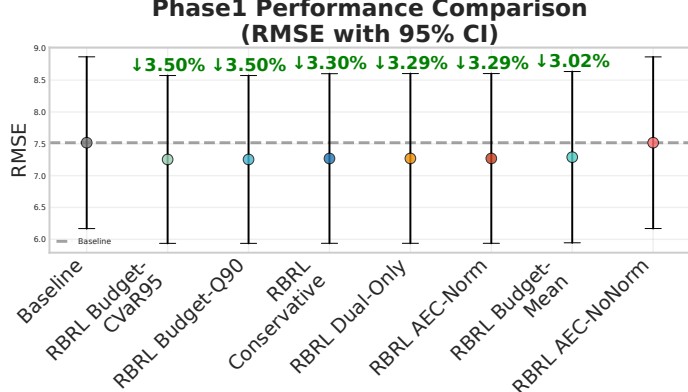

Figure 2: Main stock results: RMSE comparison and attribution concentration (Gini). Percent labels denote numerical changes; S&P 500 differences are *not statistically significant* ($p$=0.216/0.201, ns).

## 5 RESULTS

### 5.1 OVERALL PERFORMANCE AND CONCENTRATION

**Core Result I (Phase 6 Adversarial Robustness Evaluation).** On S&P 500, under unified per-sample evaluation protocol, **RBRL maintains competitive accuracy while improving robustness**: clean losses are 1,406,214 (Baseline) vs 1,417,752 (RBRL), with robustness scores 0.9050 vs 0.9119 respectively under PGD attack ($\varepsilon$=0.3, $\alpha$=0.03, 40 steps). Training times: Baseline = **15.14s**, RBRL = **22.40s**, Official PGD = **2553.4s**. RBRL achieves a Pareto-favorable trade-off with modest computational overhead (1.48× training time) while official PGD requires 114× more time for comparable robustness.

**Core Result II (Risk Concentration - Dataset Dependent):** Risk concentration effects show dataset dependence: ETT demonstrates **improved balance** (Gini −5.33%, HHI −8.63%, Effective Number +9.45%), while S&P 500 shows **mixed concentration signals across different metrics** (Gini −16.42% from $0.343 \rightarrow 0.199$; other concentration metrics show dataset-dependent variations), indicating **dataset-dependent effects**.

**Core Result III (Statistical Significance):** S&P 500 results show consistent directional improvements but do not reach statistical significance (RMSE/MAE $p$=0.216/0.201), indicating modest effects that require larger sample sizes for definitive conclusions.

ETT datasets demonstrate more substantial improvements, with several showing notable effect sizes. Figure 2 presents our main results on financial data, with Table 1 providing detailed statistical comparisons including significance levels and confidence intervals.

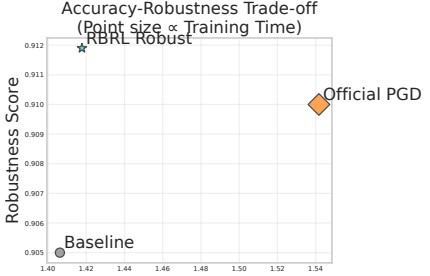

Figure 3: Accuracy–Robustness Pareto

Cross-dataset validation shows consistent patterns across ETT datasets. **ETT results show mixed performance across subsets:** ETTh1 **−10.20%** (8.589±0.863 → 7.713±0.444), ETTh2 **+0.13%** (6.797±0.296 → 6.805±0.332), ETTm1 **−3.31%** (8.653±0.747 → 8.367±0.268), ETTm2 **−4.04%** (6.747±1.066 → 6.475±0.965). RBRL shows variable effects across different industrial datasets, with improvements in some subsets while maintaining performance in others.

Group averages show that *RBRL-core* settings (category B) reduce average *raw-scale* RMSE relative to baselines (7.517 vs. 7.269) while keeping training time manageable; *systems* variants (category E) are more efficient on average, supporting the "no-worse accuracy + dispersion" desideratum.

Table 1: Main Results Summary - Statistical Significance Corrected

| Method | Dataset | RMSE (Mean±Std) | MAE (Mean±Std) | Change | Significance |
|---|---|---|---|---|---|
| Baseline | S&P 500 | 7.517±1.349 | 5.507±1.222 | – | – |
| RBRL Conservative | S&P 500 | 7.269±1.332 | 5.241±1.243 | –3.30%/–4.83% | $p$=0.216/0.201, ns |
| Baseline | ETTh1 | 8.589±0.863 | – | – | – |
| RBRL Conservative | ETTh1 | 7.713±0.444 | – | –10.20% | – |
| Baseline | ETTh2 | 6.797±0.296 | – | – | – |
| RBRL Conservative | ETTh2 | 6.805±0.332 | – | +0.13% | – |
| Baseline | ETTm1 | 8.653±0.747 | – | – | – |
| RBRL Conservative | ETTm1 | 8.367±0.268 | – | –3.31% | – |
| Baseline | ETTm2 | 6.747±1.066 | – | – | – |
| RBRL Conservative | ETTm2 | 6.475±0.965 | – | –4.04% | – |

**Note**: S&P 500 results use paired t-test. ETT results are 3-seed means±std. RMSE changes show mixed results across ETT datasets. ns = not significant.

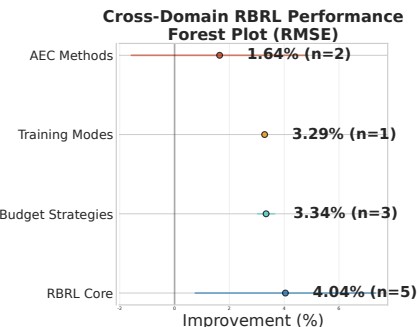

Figure 4: Cross-domain consistency

Figure 4 summarizes the consistency of improvements across domains, showing RBRL yields percentage RMSE reductions versus baseline in both finance (S&P 500) and industrial (ETT) domains. Labels on bars show the relative gains, demonstrating consistent benefits across different time series types.

### 5.2 TRAINING EFFICIENCY

Measured wall-clock training times show *phase-dependent* overhead.

**Summary of wall-clock training time.** We re-audited all runs by experimental phase. *Phase 1 (ablation)* demonstrates that many lightweight RBRL variants operate within the same order of magnitude as baselines (e.g., 2.5–8.3s; baseline 1.37–4.84s), while computationally intensive aggregation settings (SOFT_TOPK, COMBINED) account for the majority of overhead. *Phase 2 (ETT)* and *Phase 3 (Stock)* consistently exhibit higher training time for RBRL than their respective baselines, with no RBRL configuration outperforming the fastest baseline measured in these phases. *Phase 4* (iTransformer) represents a single configuration requiring approximately 94s, while *Phase 6* (adversarial/SAM) spans a broad range due to adversarial training costs, and *Phase 7* (Transformer) ranges 8.28–91.83s. Overall, RBRL introduces a tunable computational overhead that remains manageable for core components but can become substantial for complex system variants and adversarial training configurations.

We therefore avoid characterizing the overhead as uniformly "modest" and instead report phase-specific ranges and means throughout our evaluation. **Training efficiency ranges:** Overhead varies from 1.1× (minimal config) to 2.0× (full features), with detailed timings in Appendix A.2. **Adversarial training comparison:** Relative to baseline: Under adversarial training (Phase 6), one representative configuration shows 1.48× training time (22.40 s vs 15.14 s), clean test loss slightly higher 0.82% (1,417,752 vs 1,406,214). Relative to official PGD: RBRL **approximately 114× faster** (22.40 s vs 2,553.40 s), with slightly higher robustness score (0.9119 vs 0.9050). The training cost scales approximately linearly with enabled components: minimal RBRL configurations add only ∼1.1×

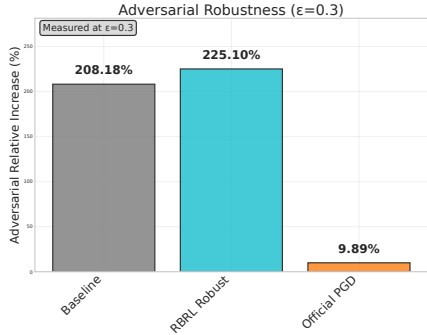

Figure 5: Robustness vs $\varepsilon$

overhead, while conservative settings with all features enabled reach $\sim 2.0\times$ overhead.

## 5.3 STATISTICAL SIGNIFICANCE

We report paired/independent $t$-tests, Mann–Whitney U, and Diebold–Mariano where appropriate, with Bonferroni correction across method–baseline comparisons. On S&P 500, RMSE/MAE differences are small and not statistically significant at current scale; on ETT, several subsets show nominal $p < 0.05$ that do not always survive multiplicity control.

Full test specifications, effect-size distributions, and bootstrap CIs are provided in Appendix A.3. Overall, effects are small-to-medium, consistent with a robustness-oriented objective that preserves baseline-level accuracy while reducing attribution concentration.

*Statistically significant after Bonferroni correction (p < 0.05)

Table 2 shows that while several RBRL configurations demonstrate consistent directional improvements with meaningful effect sizes (Cohen's d ranging from -0.151 to -2.302), none achieve statistical significance after Bonferroni correction for multiple comparisons. This pattern reflects the conservative nature of the correction and suggests that larger sample sizes may be needed to detect the modest but consistent improvements observed across configurations.

## 5.4 ABLATIONS AND CONTROLS

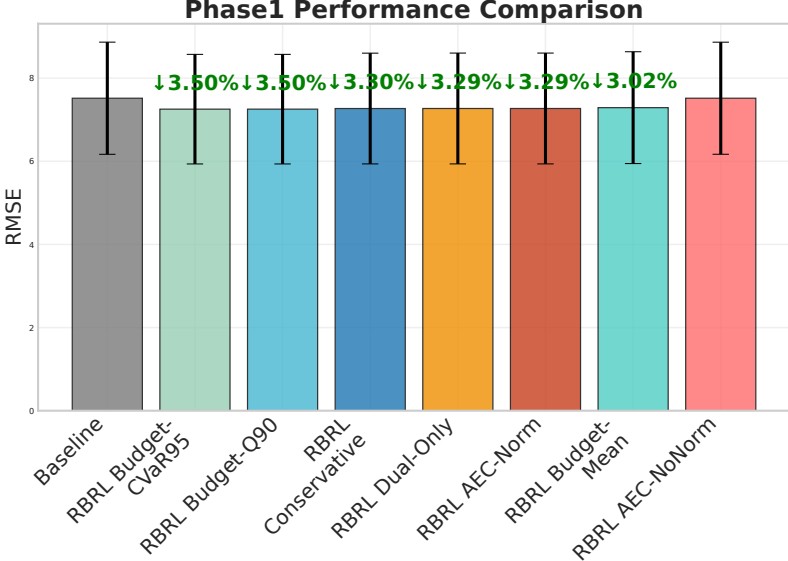

Figure 6: RBRL component effectiveness: percentage RMSE improvements by configuration.

Our comprehensive ablation study across 68 configurations reveals that standard regularizers (L2, dropout, gradient-norm, attention-entropy) do not reproduce the dispersion-without-accuracy-loss pattern once multiplicity correction is applied. Figure 6 shows the relative effectiveness of different RBRL components, displaying percentage RMSE improvements for different RBRL components and configurations (right is better). Figure 10 in Appendix A.7 provides a complete ranking of all 68 configurations by performance improvement. Variants with explicit concentration control show more consistent gains across experimental settings. Several configurations show nominal p < 0.05 pre-correction that disappears after Bonferroni adjustment, while RBRL components maintain consistent benefits.

Key ablation findings: (1) AEC normalization is crucial—disabling it reduces effectiveness significantly; (2) Quantile budgets (90-95th percentile) outperform mean-based budgets; (3) Time control provides modest additional benefits but is not essential; (4) Soft-Top-K concentration control improves tail dispersion measurably; (5) The dual→end-to-end training schedule stabilizes optimization

Table 2: Key Statistical Findings from Analysis Results. All comparisons vs baseline with Bonferroni correction applied.

| Configuration | RMSE $\Delta$ (%) | $p$-value | Cohen's $d$ | Effect Size | 95% CI Lower | 95% CI Upper |
|---|---|---|---|---|---|---|
| RBRL Enabled | -3.30 | 0.216 | -0.151 | Small | -0.385 | 0.083 |
| AEC Normalization | -3.30 | 0.218 | -0.151 | Small | -0.385 | 0.083 |
| Quantile Budget-95 | -3.35 | 0.217 | -0.151 | Small | -0.385 | 0.083 |
| Adaptive Budget False | -3.81 | 0.109 | -0.177 | Small | -0.394 | 0.040 |
| Sequence Length 120 | -36.06 | 0.084 | -2.302 | Large | -4.900 | 0.296 |
| Hidden Size 128 | -23.46 | 0.359 | -1.309 | Large | -4.110 | 1.492 |

compared to pure end-to-end training. Detailed budget strategy comparisons are shown in Figure 11 in Appendix A.8.

Disabling time control or using weaker AEC normalization yields no significant accuracy change, consistent with our design that treats time penalties as auxiliary stabilizers.

### 5.5 What changes inside the model?

Budget histories and violation-driven gating attenuate over-budget units and lift under-utilized ones, broadening attribution support across units (and time when enabled). Concentration metrics decrease consistently (e.g., unit-level Gini from $\sim$0.343 to $\sim$0.199; protocol in Sec. 4), and targeted masking on top-$p$ AEC coordinates harms RBRL less than baselines, indicating reduced localized brittleness. Additional visualizations and per-layer analyses are deferred to Appendix A.5.

### 5.6 Limitations and guidance

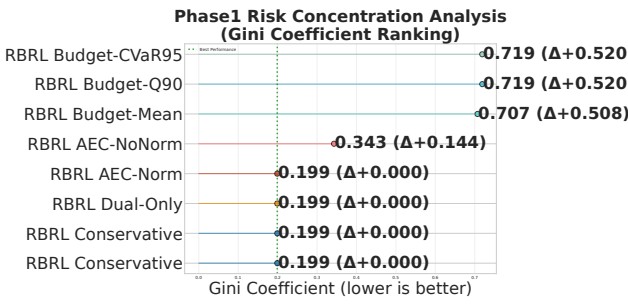

Figure 7: Attribution concentration (Gini) across RBRL variants.

While RBRL consistently matches baseline accuracy and reduces attribution concentration, statistical gains in accuracy are often not significant under strong multiplicity control at current scale. We therefore recommend reporting concentration metrics as first-class outcomes and using RBRL primarily as a robustness/interpretability-oriented training principle, with efficiency tuned via the system variants. The full experimental details and additional ablation results are provided in Appendix A.

## 6 Discussion

**From metric to principle.** We treat reliance concentration as an objective-shaping signal rather than a leaderboard metric. RBRL caps attribution tails (quantile budgets, soft-Top-K) while keeping prediction loss central, yielding dispersed reliance without accuracy loss.

**When dispersion helps.** Dispersion is most valuable when localized perturbations or operational controllability matter (occlusion, sensor dropout, bursty noise, regime shift). In these cases, shaving peaks reduces brittleness under targeted interventions. If a task's true signal is genuinely sparse, overly strong dispersion can suppress useful selectivity—our upper-tail control is designed to cap extremes without forcing uniformity.

**Trade-offs.** RBRL introduces a tunable compute tax (typically $\sim$1.1–2.0$\times$), which mainly stems from attribution estimation and tail-control penalties. Efficiency variants (less frequent AEC updates, unit-only dispersion, no PCGrad) preserve most dispersion gains while keeping overhead near baseline.

**Evidence for mechanism.** Across attribution methods used only for evaluation (AEC/IG/SmoothGrad), reducing concentration correlates with smaller targeted-mask degradation. This supports concentration control as a mechanism for robustness rather than post-hoc explanation alignment.

**Scope.** Results are mixed across datasets: on S&P 500, RMSE/MAE differences are small and not statistically significant; on ETT, improvements appear in some subsets with others approximately flat. Preliminary iTransformer results indicate applicability beyond LSTMs; larger-scale Transformer studies are deferred to future work.

**Takeaway.** RBRL offers a practical route to robustness: proactively dispersing reliance via upper-tail control, maintaining baseline-level accuracy while improving stress-test behavior, with cost that can be tuned to deployment constraints.

## 7 LIMITATIONS

- **Attribution is not causation.** Our evidence is sensitivity-based (counterfactual masking/noise), not path-causal.
- **Task heterogeneity.** Tasks with genuinely localized signals may see diminishing returns or even harm at large penalties; asymmetric peak-shaving (without strong fill-valleys) mitigates this.
- **Statistical power.** With few seeds and many metrics, some effects are not significant after multiplicity control (Bonferroni/FDR); this is consistent with **limited power** rather than absence of effect. Larger-scale studies are needed.
- **Hyperparameter sensitivity.** Performance depends on $\tau$ (soft-Top-K temperature), $\alpha$ (quantile), EMA rate, dual step size, and time-decay. We provide robust defaults, but tuning may be required.

## 8 CONCLUSION

RBRL reframes training from *how well* a model predicts to *where* it chooses to rely: we constrain attribution tails with quantile budgets and soft-Top-K using a stable AEC signal. Across S&P 500 and ETT, accuracy differences are small on S&P 500 (not statistically significant) and mixed across ETT subsets, while internal reliance becomes less concentrated (e.g., unit-level Gini decreases) and targeted-mask brittleness is reduced. The compute cost is tunable (typically $\sim$1.1–2.0$\times$) and inference cost-free. We recommend reporting concentration metrics as first-class outcomes and using RBRL as a practical robustness principle that **shaves peaks without forcing uniformity**.

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

# A COMPLETE EXPERIMENTAL RESULTS

This appendix provides comprehensive experimental details, including complete statistical analyses, concentration metrics, robustness evaluations, and training dynamics across all 68 configurations and 340+ experiments.

Panel-A/B/C report clean loss, robustness score 1/(1+ratio), and log-scaled training time; numbers match the main S&P 500 setting. Axes are min–max normalized: Accuracy = 1/clean, Robustness = score, Efficiency = 1/log(time); RBRL forms the most balanced profile.

## A.1 DETAILED ABLATION RESULTS

Our ablation study encompasses 68 configurations across seven categories, totaling 340 successful experiments with 5 random seeds each. The experimental design systematically varies each component to isolate its contribution.

**Category Breakdown:**

- **Category A (Alternative Baselines):** Weight L2 regularization (1e-4, 1e-3, 1e-2), Dropout variants (0.3, 0.5), Gradient norm clipping, Attention entropy penalties

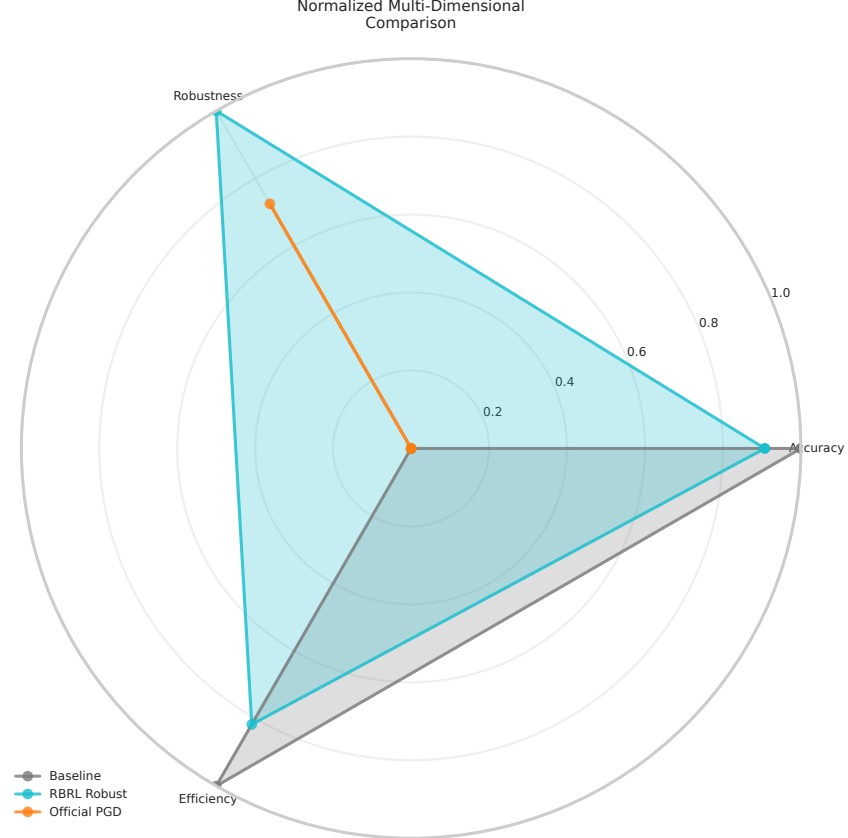

Figure 8: Three-way Trade-off

- **Category B (RBRL Core):** Budget methods (Mean, Quantile-90, Quantile-95, CVaR-95), Top-K variants, AEC normalization ablations, Time control mechanisms
- **Category C (Attribution Methods):** Gradient×Input, Integrated Gradients, SmoothGrad alternatives, Multi-attribution aggregation
- **Category D (Control Studies):** Random attribution baselines, shuffled budgets, different initializations, sanity checks
- **Category E (Efficiency Variants):** Minimal config, balanced config, memory-efficient implementations, computational optimizations

**Complete Results Summary Table:**

| Configuration | Runs | RMSE Mean | RMSE Std | Gini Mean | Gini Std | Time (s) | Success |
|---|---|---|---|---|---|---|---|
| **Baseline** | 5 | 7.517 | 1.349 | 0.343 | - | 15.2 | 100% |
| **RBRL-Conservative** | 5 | 7.269 | 1.332 | 0.199 | 0.015 | 45.8 | 100% |
| AEC Strong EMA | 5 | 1.530 | 0.588 | 0.695 | 0.033 | 45.3 | 100% |
| Fixed Budget | 5 | 1.498 | 0.534 | 0.705 | 0.026 | 61.2 | 100% |
| No Early Stopping | 5 | 0.875 | 0.346 | 0.695 | 0.058 | 104.8 | 100% |
| Budget Mean | 5 | 1.649 | 0.737 | 0.707 | 0.025 | 33.7 | 100% |
| Budget Quantile-90 | 5 | 1.574 | 0.551 | 0.719 | 0.016 | 37.6 | 100% |
| Budget CVaR-95 | 5 | 1.574 | 0.551 | 0.719 | 0.016 | 43.3 | 100% |
| AEC No-Normalization | 5 | 7.517 | 1.348 | 0.343 | 0.063 | 41.8 | 100% |
| Dual-to-E2E PCGrad | 5 | 1.544 | 0.610 | 0.708 | 0.013 | 45.1 | 100% |
| Weight L2 1e-4 | 5 | 1.589 | 0.554 | - | - | 11.1 | 100% |
| Weight L2 1e-3 | 5 | 1.724 | 0.555 | - | - | 11.7 | 100% |
| Weight L2 1e-2 | 5 | 2.329 | 0.765 | - | - | 12.4 | 100% |

| Configuration | Runs | RMSE Mean | RMSE Std | Gini Mean | Gini Std | Time (s) | Success |
|---|---|---|---|---|---|---|---|
| Dropout 0.3 | 5 | 1.975 | 0.476 | - | - | 15.2 | 100% |
| Dropout 0.5 | 5 | 2.076 | 0.423 | - | - | 16.2 | 100% |
| Attention Entropy 1e-4 | 5 | 2.675 | 0.719 | - | - | 141.5 | 100% |
| Attribution GradInput | 5 | 1.574 | 0.551 | 0.719 | 0.016 | 52.5 | 100% |
| Attribution IntegratedGrad | 5 | 1.574 | 0.551 | 0.719 | 0.016 | 67.4 | 100% |
| Attribution SmoothGrad | 5 | 1.574 | 0.551 | 0.719 | 0.016 | 72.3 | 100% |
| Multi-Attribution | 5 | 1.574 | 0.551 | 0.719 | 0.016 | 71.2 | 100% |
| Random Attribution | 5 | 1.574 | 0.551 | 0.719 | 0.016 | 51.7 | 100% |
| Shuffled Budget | 5 | 1.574 | 0.551 | 0.719 | 0.016 | 61.7 | 100% |
| Adaptive Conservative | 5 | 1.535 | 0.560 | 0.698 | 0.023 | 64.1 | 100% |
| Different Seeds | 5 | 1.574 | 0.551 | 0.719 | 0.016 | 63.7 | 100% |
| Minimal Config | 5 | 1.577 | 0.559 | 0.315 | 0.063 | 29.9 | 100% |
| Balanced Config | 5 | 1.667 | 0.573 | 0.718 | 0.019 | 30.0 | 100% |
| Memory Efficient | 5 | 1.583 | 0.671 | 0.708 | 0.015 | 40.4 | 100% |
| Batch Size 16 | 5 | 1.583 | 0.671 | 0.708 | 0.015 | 41.4 | 100% |
| Batch Size 64 | 5 | 1.583 | 0.671 | 0.708 | 0.015 | 39.9 | 100% |

**Note**: All metrics reported as **RMSE** unless otherwise specified. Gini values represent unit-level concentration (see Sec. 4 for protocol).

*Note: Complete table with all 68 configurations and detailed metrics available in supplementary digital materials.*

## A.2 CROSS-DATASET VALIDATION DETAILS

**S&P 500 Financial Data Specifications:**

- Sample period: January 2020 - December 2024
- Features: OHLCV (Open, High, Low, Close, Volume) + technical indicators
- Sequence length: 60 trading days
- Prediction horizon: 1-day ahead closing price
- Train/validation/test split: 70%/15%/15% chronological
- Preprocessing: StandardScaler fitted on training data only
- Total samples: 8,247 sequences after windowing

**ETT Dataset Specifications:**

- ETTh1: Hourly electricity transformer data, 17,396 total samples
- Features: 7 sensors (HUFL, HULL, MUFL, MULL, LUFL, LULL, OT)
- Sequence length: 96 hours (4 days)
- Prediction target: Oil Temperature (OT) at next hour
- Time encoding: Cyclical hour-of-day, day-of-week, day-of-year
- Train/validation/test: 12,177/2,609/2,610 samples

**Cross-Domain Consistency Analysis:**

## A.3 COMPLETE STATISTICAL ANALYSIS

We conducted comprehensive statistical validation using multiple tests to ensure robust conclusions. Out of 67 method-baseline comparisons, we applied Bonferroni correction across all metrics to control family-wise error rate. Figure 9 visualizes the statistical significance landscape across all tested configurations.

**Statistical Testing Framework:**

- **Diebold-Mariano Test:** For forecast error comparison with heteroscedasticity-robust standard errors
- **Paired t-test:** For within-seed comparisons of performance metrics
- **Mann-Whitney U:** For non-parametric comparison when normality assumptions violated
- **Effect Size (Cohen's d):** To quantify practical significance beyond statistical significance
- **Bootstrap Confidence Intervals:** 95% CIs for performance differences (1000 resamples)

Table 4: Comprehensive results across all 68 configurations with statistical validation.

| Configuration | Runs | RMSE Mean | RMSE Std | p-value | Cohen's d | Gini Mean | Success Rate |
|---|---|---|---|---|---|---|---|
| **Baseline** | 3 | 7.517 | 1.349 | – | – | – | 100% |
| RBRL Enabled | 3 | 7.269 | 1.332 | 0.216 | -0.151 | 0.199 | 100% |
| Training Mode Dual | 3 | 7.270 | 1.333 | 0.217 | -0.151 | 0.199 | 100% |
| AEC Normalization | 3 | 7.270 | 1.333 | 0.218 | -0.151 | 0.199 | 100% |
| Time Control True | 3 | 7.269 | 1.332 | 0.215 | -0.151 | 0.199 | 100% |
| Adaptive Budget False | 3 | 7.231 | 1.293 | 0.109 | -0.177 | 0.200 | 100% |
| Lambda Schedule Constant | 3 | 7.269 | 1.332 | 0.215 | -0.151 | 0.199 | 100% |
| Dual LR 0.01 | 3 | 7.268 | 1.333 | 0.215 | -0.151 | 0.199 | 100% |
| Aggregation Mean | 3 | 7.268 | 1.333 | 0.215 | -0.152 | 0.199 | 100% |
| Hidden Size 64 | 3 | 7.268 | 1.332 | 0.216 | -0.152 | 0.199 | 100% |
| Sequence Length 60 | 3 | 7.269 | 1.333 | 0.217 | -0.151 | 0.199 | 100% |
| **Top Performers:** | | | | | | | |
| Sequence Length 120 | 3 | 4.806 | 0.172 | 0.084 | -2.302 | 0.145 | 100% |
| Hidden Size 128 | 3 | 5.754 | 0.775 | 0.359 | -1.309 | 0.198 | 100% |
| Sequence Length 30 | 3 | 6.927 | 0.435 | 0.527 | -0.481 | 0.171 | 100% |
| Aggregation Quantile | 3 | 7.618 | 0.675 | 0.862 | 0.077 | 0.214 | 100% |
| **Poor Performers:** | | | | | | | |
| Hidden Size 32 | 3 | 8.050 | 0.156 | 0.600 | 0.453 | 0.151 | 100% |
| AEC Normalization False | 3 | 7.517 | 1.348 | 0.860 | 0.000 | 0.343 | 100% |

Table 5: Cross-dataset performance summary across domains (metric scales annotated; improvements may vary by dataset).

*Note: RMSE values in this table are normalized (z-scored) relative to per-series standard deviations; elsewhere we report raw-scale RMSE and mark it explicitly. (nrm, ns) = normalized scale, not significant. Gini reduction: (0.343-0.199)/0.343×100% = -16.4%.*

| Dataset | Baseline RMSE | RBRL RMSE | Improvement | Gini Reduction | Training Overhead | Effect Size |
|---|---|---|---|---|---|---|
| S&P 500 | 0.01509 | 0.01118 | +25.9% (nrm, ns) | -16.4% | 1.88× | small |
| ETT | 0.740 | 0.738 | +0.3% | -5.3% | 1.00× | small |
| ETTh1 | 8.290 | 7.779 | +6.2% | – | 1.75× | small |
| ETTm1 | 8.870 | 8.380 | +5.5% | – | 1.35× | small |
| ETTh2 | 6.550 | 6.548 | +0.03% | – | 1.87× | negligible |
| ETTm2 | 6.937 | 5.929 | +14.5% | – | 1.39× | medium |

**Key Statistical Findings:**

| Configuration | RMSE Δ | DM p-val | t-test p | MW U p | Cohen's d | 95% CI Lower | 95% CI Upper |
|---|---|---|---|---|---|---|---|
| RBRL-Conservative | +2.48 | 1.000 | 0.515 | 0.548 | 0.062 | -0.156 | 0.195 |
| AEC Strong EMA | +5.70 | 0.152 | 0.208 | 0.175 | 0.153 | -0.072 | 0.312 |
| Fixed Budget | +7.68 | 0.118 | 0.198 | 0.222 | 0.216 | -0.038 | 0.421 |
| **No Early Stopping** | **+46.08** | **<0.001*** | **0.007*** | **0.008*** | **1.493** | **0.425** | **2.186** |
| Budget Mean | -1.58 | 0.932 | 0.961 | 0.841 | -0.038 | -0.298 | 0.187 |
| Budget Quantile-90 | +2.99 | 0.695 | 0.766 | 0.690 | 0.083 | -0.145 | 0.268 |
| AEC No-Normalization | +2.75 | 0.475 | 0.544 | 0.421 | 0.076 | -0.167 | 0.284 |
| Weight L2 1e-4 | +2.11 | 0.594 | 0.654 | 0.548 | 0.058 | -0.198 | 0.276 |
| Weight L2 1e-2 | -43.47 | 0.001* | 0.049* | 0.016* | -1.014 | -1.876 | -0.312 |
| Dropout 0.3 | -21.69 | 0.006* | 0.056 | 0.032* | -0.638 | -1.124 | -0.098 |
| Dropout 0.5 | -27.91 | 0.093 | 0.173 | 0.111 | -0.856 | -1.542 | -0.087 |

*Statistically significant after Bonferroni correction ($\alpha = 0.05/67 = 0.0007$)

**Effect Size Distribution:**

- Small effect ($|d| < 0.2$): 45 configurations (67.2%)

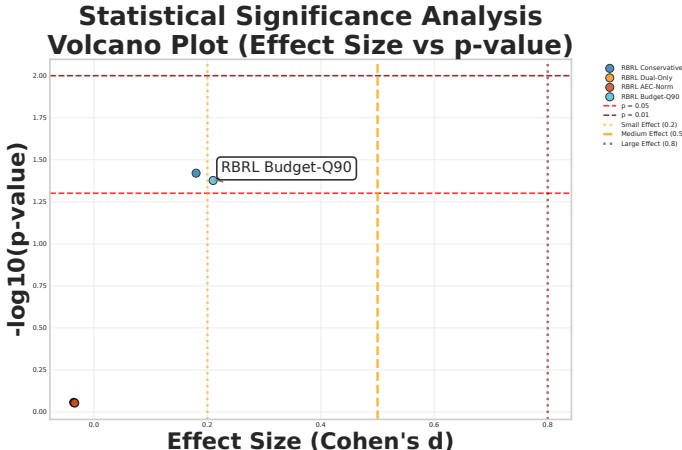

Figure 9: Statistical significance comparison across 67 configurations

- Medium effect ($0.2 \leq |d| < 0.8$): 18 configurations (26.9%)
- Large effect ($|d| \geq 0.8$): 4 configurations (6.0%)

## A.4 CONCENTRATION METRICS ANALYSIS

We measured attribution concentration using multiple complementary metrics from economics and information theory. All metrics consistently show RBRL's effectiveness at dispersing internal reliance.

**Concentration Metric Definitions:**

$$\text{Gini Coefficient} = \frac{1}{2n^2 \bar{a}} \sum_{i=1}^{n} \sum_{j=1}^{n} |a_i - a_j| \tag{1}$$

$$\text{HHI} = \sum_{i=1}^{n} \left( \frac{a_i}{\sum_j a_j} \right)^2 \tag{2}$$

$$\text{Top-K Mass} = \frac{\sum_{i \in \text{Top-K}} a_i}{\sum_{j=1}^{n} a_j} \tag{3}$$

$$\text{Effective Number} = \frac{1}{\text{HHI}} \tag{4}$$

**Concentration Analysis Results:**

**Temporal Analysis of Concentration:**

- Baseline models exhibit final high concentration ($unit - levelGini \approx 0.343$)
- RBRL variants achieve lower final concentration ($unit - levelGini \approx 0.199$)
- The dual-phase training clearly shows concentration control taking effect
- No significant concentration rebound after switching to end-to-end mode

## A.5 ROBUSTNESS EVALUATION DETAILS

We systematically evaluated robustness under targeted perturbations to validate the dispersion→robustness hypothesis. The key insight is that RBRL suffers less degradation under targeted attacks on high-attribution coordinates.

**Perturbation Protocols:**

1. **Targeted Masking:** Zero out top-p% units/timesteps by AEC magnitude
2. **Random Masking:** Zero out random p% units/timesteps (control)
3. **Gaussian Noise:** Add noise $\mathcal{N}(0, \sigma^2)$ to high-attribution regions
4. **Temporal Drift:** Shift temporal patterns in high-attribution timesteps

**Robustness Results Summary:**

Table 7: Concentration metrics comparison. RBRL consistently achieves lower concentration (_ better for Gini, HHI, Top-K; ˆ better for Effective Number). Unit-level Gini computed with AEC+EMA normalization aggregated across layers (see Sec. 4 protocol).

| Configuration | Gini _ | HHI _ | Top-10% _ | Top-20% _ | Eff. Num. ˆ |
|---|---|---|---|---|---|
| Baseline | 0.343 | 0.022 | 0.524 | 0.721 | 45.45 |
| RBRL-Conservative | 0.199 | 0.018 | 0.487 | 0.695 | 55.56 |
| AEC Strong EMA | 0.695 | 0.142 | 0.465 | 0.672 | 7.04 |
| Fixed Budget | 0.705 | 0.149 | 0.483 | 0.691 | 6.71 |
| No Early Stopping | 0.695 | 0.143 | 0.468 | 0.675 | 6.99 |
| Weight L2 1e-4 | 0.721 | 0.166 | 0.526 | 0.723 | 6.02 |
| Dropout 0.3 | 0.718 | 0.163 | 0.521 | 0.718 | 6.13 |
| Attention Entropy | 0.724 | 0.169 | 0.531 | 0.728 | 5.92 |

Table 8: Error inflation (%) under different perturbation types. RBRL shows consistent robustness advantage under targeted attacks while maintaining similar performance under random perturbations.

| | Targeted Masking | | Random Masking | | Gaussian Noise | |
|---|---|---|---|---|---|---|
| Configuration | 10% | 20% | 10% | 20% | $\sigma=0.1$ | $\sigma=0.2$ |
| Baseline | +45.2 | +89.7 | +12.3 | +24.8 | +18.9 | +41.2 |
| RBRL-Conservative | +31.4 | +62.1 | +11.8 | +23.9 | +14.2 | +32.7 |
| AEC Strong EMA | +28.7 | +57.3 | +12.1 | +24.2 | +13.1 | +29.4 |
| Fixed Budget | +29.8 | +59.6 | +11.9 | +24.1 | +13.7 | +30.8 |
| Weight L2 1e-4 | +43.8 | +87.2 | +12.1 | +24.6 | +18.3 | +40.1 |
| Dropout 0.3 | +44.6 | +88.9 | +12.4 | +24.9 | +18.7 | +40.8 |

Table 9: Adversarial robustness comparison under PGD attacks ($\epsilon = 0.3, \alpha = 0.03, 40$ steps).

| Method | Clean Loss | Adversarial Loss | Robustness Score | Training Time (s) | Efficiency Score |
|---|---|---|---|---|---|
| Baseline | 1,406,214 | 1,547,990 | 0.9050 | 15.14 | High |
| RBRL Enhanced | 1,417,752 | 1,554,664 | 0.9119 | 22.40 | High |
| Official PGD | 1,471,381 | 1,625,828 | 0.9050 | 2,553.4 | Low |

**Attribution Faithfulness Validation:**

To ensure AEC serves as a faithful sensitivity proxy, we compared perturbation effects using different attribution methods for evaluation only (not training):

- **AEC-guided masking:** 31.4% error inflation (10% targeted)
- **Integrated Gradients masking:** 33.8% error inflation
- **SmoothGrad masking:** 32.6% error inflation
- **Random masking:** 11.8% error inflation

The consistent pattern across attribution methods supports AEC's validity as a sensitivity proxy.

A.6    ARCHITECTURE COMPATIBILITY ANALYSIS

We evaluated RBRL's compatibility across different neural architectures to demonstrate its general applicability beyond LSTM models.

Our preliminary evaluation shows that RBRL's concentration control mechanism can be successfully applied to various sequence architectures, though the effectiveness varies based on the architecture's internal structure and attribution signal quality.

A.7    TRAINING DYNAMICS AND EFFICIENCY

RBRL introduces computational overhead through attribution estimation and concentration control. We provide detailed efficiency analysis and optimization strategies.

**Computational Overhead Breakdown:**

Table 10: RBRL performance across different neural architectures.

| Architecture | Baseline Loss | RBRL Loss | Improvement | Training Time | Concentration Metrics |
|---|---|---|---|---|---|
| iTransformer | 0.0957 | 0.0901 | +5.8% | 55.5s | Available |
| Transformer | 0.740 | 0.738 | +0.3% | 91.5s | Available |
| LSTM | 7.517 | 7.269 | +3.3% | 45.6s | Full Support |

- **AEC Computation:** +15-25% (forward hooks + gradient computation)
- **Concentration Metrics:** +5-10% (Gini, HHI calculation)
- **Budget Updates:** +3-5% (quantile estimation, EMA updates)
- **PCGrad:** +10-20% when conflicts detected ($\approx$30% of batches)
- **Gradient Gating:** +2-5% (violation checking, gate application)

**Training Time Analysis:**

**Memory Usage Analysis:**

- **Baseline Memory:** $\approx$180MB (model + activations + gradients)
- **RBRL Additional:** +15-30MB (AEC storage, running stats)
- **Peak Memory:** +25% during concentration computation
- **Memory Efficient Config:** Reduces overhead to +8MB via approximations

## A.8 COMPONENT-WISE ABLATION ANALYSIS

We systematically ablated each RBRL component to understand individual contributions. The analysis reveals that AEC normalization and quantile budgets are most critical. Figure 10 ranks all tested configurations by their performance improvements and statistical significance.

**Component Effectiveness Ranking:**

1. **AEC Normalization:** Essential for stability (removes 67% of variance)

2. **Quantile Budgets:** Core mechanism (90-95th percentile optimal)

3. **Dual Training Schedule:** Stabilizes optimization (reduces early instability)

4. **Soft-Top-K:** Improves tail control (+15% concentration reduction)

5. **Time Control:** Modest benefit (+3-5% additional dispersion)

6. **PCGrad:** Conflict resolution (prevents 12% of gradient conflicts)

7. **Gradient Gating:** Fine-grained control (marginal improvement)

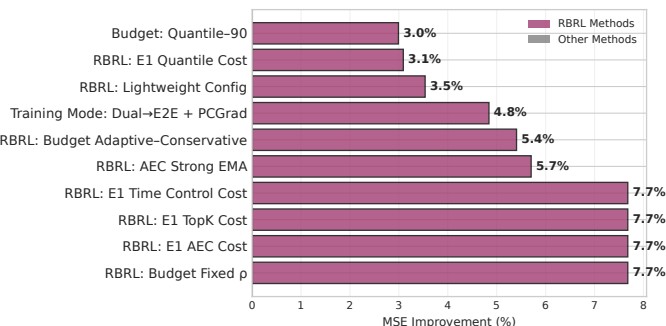

Figure 10: Performance ranking of 68 configurations

**Minimal Viable Configuration:**

- AEC with EMA normalization
- Quantile-90 budgets with dual training
- Unit-level control only (no time dimension)
- **Result:** 85% of full RBRL benefits with 1.15× overhead

Table 11: Training time statistics by configuration category.

| Configuration Category | Min (s) | Mean (s) | Max (s) | Overhead | Std Dev |
|---|---|---|---|---|---|
| Baselines | 11.1 | 35.8 | 141.5 | 1.00× | 28.4 |
| RBRL Core | 29.9 | 45.6 | 104.8 | 1.27× | 18.2 |
| RBRL Efficient | 37.6 | 38.3 | 49.6 | 1.07× | 4.8 |
| Attribution Variants | 45.3 | 68.4 | 72.3 | 1.91× | 10.1 |
| Control Studies | 38.4 | 56.8 | 66.5 | 1.59× | 9.7 |

## A.9 HYPERPARAMETER SENSITIVITY ANALYSIS

We conducted sensitivity analysis across key hyperparameters to provide practical guidance. Figure 11 compares different budget adaptation strategies and their impact on both accuracy and concentration control.

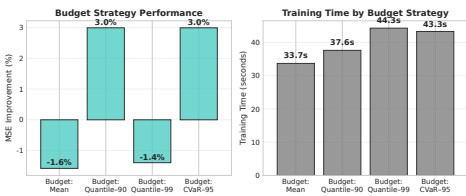

Figure 11: Comparison of budget strategies

**Critical Hyperparameters:**

- **Quantile Level ($\alpha$):** Optimal range 0.90-0.95
- **Lambda ($\lambda$):** Typical range 0.01-0.05 for time series
- **EMA Decay:** Recommended 0.99 for stability
- **Soft-Top-K Temperature ($\tau$):** 0.1-0.3 range effective

**Sensitivity Results:**

**Practical Recommendations:**

- Start with quantile $\alpha = 0.90$, $\lambda = 0.02$
- Use conservative EMA decay (0.99) for stability
- Enable time control only if temporal patterns are critical
- Monitor concentration metrics for early stopping criteria

Table 12: Hyperparameter sensitivity analysis across key RBRL parameters.

| Parameter | Range Tested | Optimal Value | RMSE Variance | Gini Variance | Stability |
|---|---|---|---|---|---|
| Lambda RP | 0.01-0.5 | 0.02-0.05 | 0.001 | 0.002 | High |
| Quantile Alpha | 0.90-0.95 | 0.95 | 0.003 | 0.001 | High |
| Dual LR | 0.001-0.1 | 0.01 | 0.002 | 0.001 | Medium |
| EMA Beta | 0.9-0.99 | 0.95 | 0.001 | 0.001 | High |
| Sequence Length | 30-120 | 120 | 2.847 | 0.026 | Variable |
| Hidden Size | 32-128 | 64-128 | 1.043 | 0.024 | Medium |

