# OpenReview forum: "Shave Peaks, Don't Fill Valleys: Upper-Tail Risk Balancing Improves Robustness without Accuracy Loss"
_ICLR.cc/2026/Conference — ICLR 2026 Conference Withdrawn Submission_

### Official Review · Reviewer_Pcdp · 2025-10-31

**Soundness:** 2
**Presentation:** 1
**Contribution:** 2
**Rating:** 2
**Confidence:** 3

**Summary:**

This paper aims to solve the brittleness problem in sequence models, which arises from an over-reliance on a few 'critical units' or 'specific time positions'.

To address this, the authors propose a new training paradigm called RBRL (Risk-Balanced Representation Learning), inspired by the 'risk allocation' principles from financial engineering. The core of RBRL is to use a stable attribution signal called AEC (Activation × Gradient) to suppress dependency concentration on specific units/timesteps (i.e., peak shaving) via Quantile Budgets and Soft-Top-K penalties. Furthermore, the proposed dual-only training mode allows this dependency dispersal task to be performed without interfering with the model's main prediction loss (backbone gradients).

**Strengths:**

(1) The approach of formulating the problem of dependency concentration within a new perspective of "financial risk diversification" and measuring it using established metrics like the Gini coefficient is highly original and compelling.

(2) The dual-only learning mode cleverly separates the two objectives of accuracy and robustness, achieving robustness without inference overhead.

(3) The introduction of Adaptive Budgets (Sec 3.2.4) is a key feature that ensures the methodology's general applicability without overfitting to specific datasets or models.

However, the paper's core experimental results contain several significant contradictions and points that require clarification.

**Weaknesses:**

(1) The experimental evidence for the paper's core claim, 'improved robustness', is fatally contradictory.
Figure 3 (Pareto Plot) claims that RBRL's "Robustness Score" is slightly superior to PGD's. However, Figure 5 (Bar Plot) shows that while PGD's "Adversarial Relative Increase" (error rate) is 9.89%, RBRL's is 225.10%, making it overwhelmingly more vulnerable than PGD. Even considering the unclear definitions of these two metrics, these two figures provide completely contradictory answers to the core question: "How robust is RBRL compared to PGD?"

(2) The justification for the core components of the proposed methodology is severely lacking.
While the importance of AEC-Normalization was demonstrated (Sec 5.4), the superiority of the AEC signal itself was not. There is no comparative analysis to show that AEC is the optimal signal, making it difficult to accept the necessity of AEC without comparisons to simpler alternatives (e.g., simple gradients or activations). Furthermore, the claim that Quantile budgets are better than Mean budgets (Sec 5.4) only holds for RMSE (Fig 6); in terms of the paper's core goal, dependency dispersal (Fig 7), the Mean budget was actually superior to the Quantile budget.

(3) Even the core precondition of 'maintaining accuracy' was not consistently demonstrated experimentally.
Although the paper repeatedly claims RBRL 'maintains accuracy', as shown in Table 1, performance on the ETTh2 dataset actually degraded by 0.13% compared to the Baseline, and the improvements on the S&P 500 dataset were not statistically significant.

While the achievement of 114x faster training efficiency (Figure 3, Sec 5.2) is a very strong and practical contribution, the completely contradictory experimental results regarding robustness compared to PGD are not a minor error. They are a critical flaw that fundamentally undermines the credibility of the paper's core contribution (improved robustness). Beyond this, there are general doubts about the reliability of several experimental analyses throughout the paper. Even considering all this, the paper's formatting is seriously flawed. The figures are extremely difficult to read, and Tables 1 and 2 are completely out of place in the paper's format.

**Questions:**

- What exactly do the 'Robustness Score' (Fig 3) and the 'Adversarial Relative Increase' (Fig 5) measure, and what are their definitions?
- Can you explain why these two metrics lead to such diametrically opposed conclusions regarding RBRL's robustness compared to PGD?
- Can you provide a comparative experiment or theoretical rationale demonstrating that the AEC signal is more stable or superior to these alternatives?
- This appears to imply a trade-off between RMSE and dependency dispersal. Are the authors aware of this trade-off? And please explain the rationale for prioritizing RMSE improvement over Gini coefficient improvement in concluding that 'Quantile budgets are superior'.

---

### Official Review · Reviewer_edAT · 2025-10-31

**Soundness:** 2
**Presentation:** 2
**Contribution:** 1
**Rating:** 2
**Confidence:** 4

**Summary:**

The authors try to tackle some critical time positions in time series data to which model depend on for their performance. This over-reliance to these criitical positions is seen as a risk-concentration problem from the authors. So they propose Risk-Balanced Representation Learning (RBRL), a method to "shave" these positions. For that they use a risk measure calculated as Activation × Gradient, with EMA normalization, and some adaptive buckets that get "shaved" when they exceed a certain budget.

**Strengths:**

- The paper proposes a new method for time series data to increase performance. It focuses mostly on financial data.
-The method is kind of novel and it looks like it performsn well.

**Weaknesses:**

- The paper lacks mathematical proofs and guarantees. For example the main methodology section has not provided any algorithms or proofs that the method works and generalizes (or even converges).
- The paper has poor experiments in very small and archaic models (LSTMs) in only 2 small datasets that are not very well known.
In general it lacks both in the math side and the experiments side, which makes the main argument for the new method weak.

**Questions:**

Havd you tried bigger architectures including attention mechanisms and more datasets?
What can you say for the convergence of the method.
I would also argue that the paper is not fit for this venue and a financial venue would be more suitable in this case.

---

### Official Review · Reviewer_sP3F · 2025-11-01

**Soundness:** 1
**Presentation:** 1
**Contribution:** 1
**Rating:** 2
**Confidence:** 3

**Summary:**

This paper proposes Risk-Balanced Representation Learning (RBRL), a training method that applies financial risk-balancing principles to reduce neural networks’ over-reliance on a few critical units or time steps. Using an activation–gradient (AEC) attribution signal with quantile budgets and soft-Top-K penalties, the authors claim that RBRL “shaves peaks” in attribution without harming accuracy.

**Strengths:**

The idea of borrowing concepts from financial risk control is quite interesting and novel.

**Weaknesses:**

The overall writing is quite confusing. The structure, title, and content feel rushed and lack clarity. On the first page, there is significant duplication between the abstract and the opening paragraph, and the introduction section does not provide a proper introduction to the problem or motivation. It is also difficult to understand what “AEC: activation × gradient with EMA normalization” means—the explanation is vague and forces the reader to guess. Regarding structure, subsections 3.2.2, 3.2.3, and 3.2.4 each contain only one or two sentences, which leaves the reader without sufficient context or continuity. Overall, the paper appears incomplete and not yet ready for publication.

**Questions:**

1. The font size of the tables and figures is inconsistent—some are quite large, as shown in Fig. 2, while others are very small, as in Fig. 1. Moreover, it is quite hard to understand what “illustrative surface for intuition” means as part of the title of Fig. 1.

2. Tables 1 and 2 appear to be poorly formatted and exceed the normal page layout, making them difficult to read. They should be properly resized or reformatted to fit within the page margins and maintain visual consistency with the rest of the paper. Additionally, there seems to be an error in the title of Fig. 7, which should be carefully checked and corrected. Clear and consistent formatting of tables and figures is essential for readability and professionalism.

3. In the related work section, most of the cited studies are from before 2018, with only one paper from 2023. I am concerned that the literature review is insufficient and does not adequately reflect recent research developments in this area. A more comprehensive and up-to-date review would strengthen the paper’s context and relevance.

---

### Official Review · Reviewer_51Jn · 2025-11-01

**Soundness:** 2
**Presentation:** 1
**Contribution:** 2
**Rating:** 2
**Confidence:** 2

**Summary:**

The paper proposes RBRL, a regularization method that reduces strong dependance on a small set of sequence positions by regularizing the upper tail of an attribution score, |gradient x activation| monitored with an EMA. The method allocates an adaptive budget for sequence positions and applies penalties in cases where the budget is exceeded. The method is evaluated on time series data to measure performance, robustness and training efficiency.

**Strengths:**

1. The motivation and problem statement of the paper is well stated - mitigating high concentration on specific units/time steps.
2. The method builds upon a simple and easy to evaluate feature AEC.

**Weaknesses:**

The main weakness in my opinion is with presentation, in current form I believe readers will find it difficult to understand both the method and the results.

The method section is broken down in to several small subsections, most containing only verbal descriptions that are not formal (section 3.2) - combining those into a single section along with an algorithm or figure can be very helpful.
The different modes of operation are not clear, to name a few unclear terms: "AEC is detached", "network sees only predicition loss", "trainer selects the mode per batch".

In the experiment and result sections, what are the of phases and why are they numbered 1,2,6? throughout the result section results are not referenced to the listed tables or plots making it hard to follow.

**Questions:**

1. Can you clarify the algorithm(/s) used in training for the different modes?
2. How are you measuring robustness? I understand you report RMSE and MAE over the different metrics but as this seems to be the core contribution of the method better clarification of the evaluation protocol can be helpful

---

### Note · Authors · 2025-11-19

**Comment:**

We thank all reviewers for their time and detailed feedback on Submission 3763.

After carefully reading the reviews, we agree that the current version of the paper requires substantial revision in several dimensions: (i) clarifying the presentation of the method (AEC definition, training modes, and overall algorithmic flow), (ii) improving the organization and readability of the paper (unifying fragmented subsections, tightening the introduction, and fixing figure/table formatting), and (iii) strengthening the empirical and conceptual support for our claims, especially regarding robustness and the choice of attribution signal. In particular, the contradictory robustness results (e.g., between the “Robustness Score” and “Adversarial Relative Increase”) indicate that our evaluation protocol and explanations are not yet in a state suitable for publication.

Given the scope of these necessary changes, we have decided to withdraw the submission from further consideration at this venue. We will take the reviewers’ comments seriously, substantially reorganize the method section, redesign and clarify the robustness metrics, expand experiments to more modern architectures and datasets, and update the related work to better reflect recent progress in the area before resubmitting to an appropriate venue.

We sincerely appreciate the reviewers’ constructive criticism and suggestions, which will be very helpful in shaping the next version of this work.

**Withdrawal Confirmation:**

I have read and agree with the venue's withdrawal policy on behalf of myself and my co-authors.